# From Unimodal to Multimodal: Scaling up Projectors to Align Modalities

## Abstract

Recent contrastive multimodal vision-language models like CLIP have demonstrated robust open-world semantic understanding, becoming the standard image backbones for vision-language applications due to their aligned latent space. However, this practice has left powerful unimodal encoders for both vision and language underutilized in multimodal applications which raises a key question: Is there a plausible way to connect unimodal backbones for zero-shot vision-language tasks? To this end, we propose a novel approach that aligns vision and language modalities using only projection layers on pretrained, frozen unimodal encoders. Our method exploits the high semantic similarity between embedding spaces of well-trained vision and language models. It involves selecting semantically similar encoders in the latent space, curating a concept-rich dataset of image-caption pairs, and training simple MLP projectors. We evaluated our approach on 12 zero-shot classification datasets and 2 image-text retrieval datasets. Our best model, utilizing DINOv2 and All-Roberta-Large text encoder, achieves 76% accuracy on ImageNet with a 20-fold reduction in data and 65 fold reduction in compute requirements. The proposed framework enhances the accessibility of model development while enabling flexible adaptation across diverse scenarios, offering an efficient approach to building multimodal models by utilizing existing unimodal architectures. Code and datasets will be released upon acceptance.

## 1 Introduction

Contrastive multimodal vision-language models have demonstrated impressive zero-shot capabilities Radford et al. (2021); Jia et al. (2021); Zhai et al. (2023). These advancements have facilitated the use of language as an API for vision tasks, with captions functioning as adaptive classes, enabling a wide range of applications. The typical objective function InfoNCE, maximizes mutual information between the global summary vector of an image and its representation. However, the use of pooling functions to create global representations poses challenges for granular tasks like segmentation, which require pixel-level features. Notably, across various vision-centric benchmarks, unimodal models such as DINOv2 significantly outperform CLIP-like models Tong et al. (2024a;b). As the field progresses, there is an increasing demand for multimodal systems that can efficiently adapt to new modalities and tasks without extensive retraining. This evolution highlights the need for more flexible and efficient approaches to multimodal learning.

While efforts have been made to develop more efficient CLIP-like models, they often compromise on performance or still demand significant resources. For instance, LiT Zhai et al. (2022) achieves comparable performance to CLIP but still requires training on 256 TPU cores with over 4 billion image-caption pairs. Smaller-scale attempts like LiLT Khan & Fu (2023) show promise in retrieval tasks but struggle with zero-shot classification accuracy.

In this paper, we propose an alternative approach to vision-language multimodal alignment that strives to address these challenges. Our method builds upon the recent findings suggesting semantic similarities between well-trained unimodal vision and language embeddings Maniparambil et al. (2024); Huh et al. (2024). By leveraging these semantic similarities, we explore the potential of creating efficient CLIP-like models by training lightweight projection layers between unimodal frozen models.

This approach has two practical benefits compared to CLIP-like models:

**Accessible development:** Training only projection heads with a dense dataset significantly reduces the computational requirements compared to full model training. This approach not only decreases the environmental impact of developing foundation models but also makes their creation more accessible to the broader research community (see Section 5.5 for detailed comparisons).

**Flexible adaptation to diverse scenarios:** By connecting unimodal encoders through lightweight projectors, our method enables the utilization of specific features from each encoder in a multimodal context. Examples include: (1) Creating multilingual vision-language models for low-resource languages by aligning DINOv2 with a multilingual text encoder using a small set of image-caption pairs in the target language (Section 5.3), as well as (2) Enabling image-paragraph retrieval by aligning visual encoders with long-context language models like BERT-large, overcoming the token limit constraints of standard CLIP models (Section 5.4).

Our framework consists of three key components:

1. **Encoder Pair Selection:** We identify semantically similar vision and language encoders using the Centered Kernel Alignment (CKA) metric.

2. **Dataset Curation:** We develop a method to collect a dense, concept-rich dataset of image-caption pairs from uncurated sources. We argue that alignment is sensitive to concept coverage, and carefully select samples that cover most of the target concepts.

3. **Lightweight Projector Training:** We train simple MLP projectors between the embedding spaces of frozen unimodal models using contrastive loss.

We evaluate our approach on zero-shot transfer to 12 different classification datasets and 2 image-text retrieval datasets. Our best projector between unimodal models, utilizing DINOv2 and All-Roberta-Large-v1, achieves **76%** accuracy on ImageNet, surpassing CLIP's performance while using approximately **20** times less data and **65** times less compute. We also demonstrate our framework's versatility across tasks like zero-shot domain transfer, multilingual classification, zero-shot semantic segmentation, and image-paragraph retrieval.

Our main contributions lie not in a specific model, but in demonstrating a new framework for vision-language alignment. In summary, we demonstrate that CLIP-like performance can be achieved by training only projection layers, using a curated, concept-rich dataset to enable efficient projector training with significantly less data and compute.

## 2 CKA vs Ease of Alignment

Previous studies Huh et al. (2024); Maniparambil et al. (2024) have shown that well-trained vision and language encoders exhibit high semantic similarity using Centered Kernel Alignment (CKA) Kornblith et al. (2019), which measures the similarity of induced graphs of concepts across different hidden representation spaces (see Section 2.1 for CKA). A layerwise analysis in Maniparambil et al. (2024) reveals that most of this similarity is concentrated in the final projection layer. Additionally, model stitching methods Lenc & Vedaldi (2015); Bansal et al. (2021); Merullo et al. (2022) demonstrate that different network regions can be stitched together using linear layers. Inspired by this, we investigate whether semantically similar encoder embedding spaces can be aligned through a simple projection transformation, using toy examples to validate the underlying concept.

### 2.1 CKA Preliminary

**Centered Kernel Alignment (CKA)** Kornblith et al. (2019) measures the similarity of induced graphs of concepts in each encoder space and can act as a guide for encoder pairs selection that are amenable to alignment as we demonstrate in section 4. We define CKA as follows: Given two sets of vectors $X$ and $Y$, CKA measures the similarity of these vectors in their respective high-dimensional feature spaces. The kernel matrices $K$ and $L$ are derived from the data sets $X$ and $Y$, respectively, and represent the inner products between the vectors in these spaces. The entries of $K$ and $L$ are computed as:

$$K_{ij} = k(\mathbf{x}_i, \mathbf{x}_j), \quad L_{ij} = l(\mathbf{y}_i, \mathbf{y}_j)$$

where $k$ and $l$ are kernel functions applied to the vectors $\mathbf{x}_i, \mathbf{x}_j \in X$ and $\mathbf{y}_i, \mathbf{y}_j \in Y$, respectively. Common choices for these kernel functions include linear kernels, where $k(\mathbf{x}_i, \mathbf{x}_j) = \mathbf{x}_i^\top \mathbf{x}_j$, or Gaussian kernels, where $k(\mathbf{x}_i, \mathbf{x}_j) = \exp(-\gamma \|\mathbf{x}_i - \mathbf{x}_j\|^2)$ for some $\gamma > 0$.

The CKA coefficient, $\text{CKA}(K, L)$, is defined as:

$$\text{CKA}(K, L) = \frac{\text{HSIC}(K, L)}{\sqrt{\text{HSIC}(K, K) \cdot \text{HSIC}(L, L)}}$$

where HSIC stands for Hilbert-Schmidt Independence Criterion Gretton et al. (2005); Ma et al. (2020), which measures the dependence between the sets of vectors. This measure is invariant to orthogonal transformations and isotropic scaling of the data, making it robust for comparing different models.

## 2.2 CKA and Ease of Alignment Toy Example

We define the ease of alignment as the minima of the training loss after convergence. We examine how CKA correlates with the minimum CLIP loss when transforming one vector set to match another using a Linear layer. Since CLIP loss lacks a closed-form solution, we applied SGD for 500 iterations per instance, recording the final loss value as the minimum. We fixed the temperature at 0.07 and the learning rate at 0.01, choosing 500 iterations because the loss value showed minimal change beyond this point.

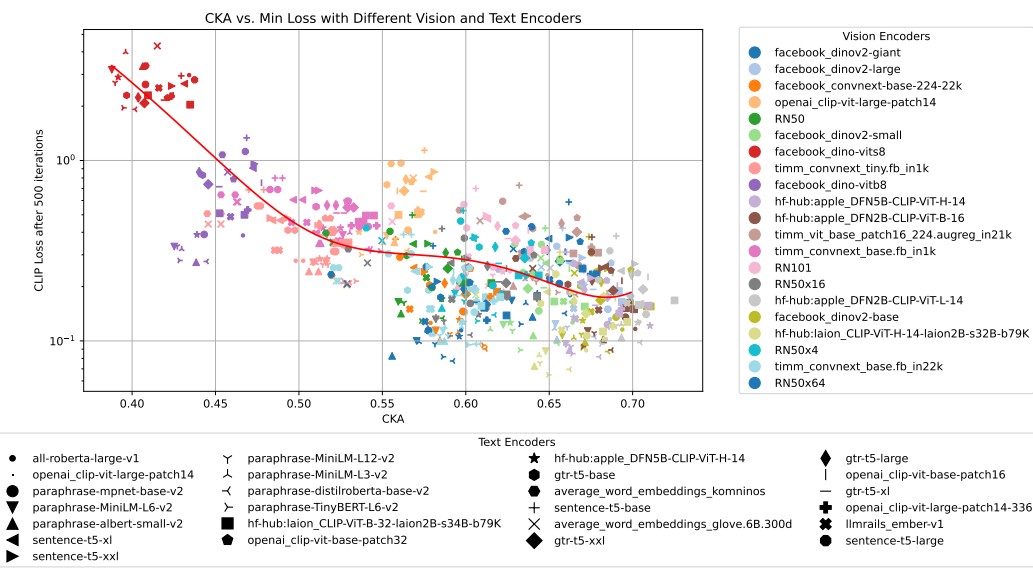

Figure 1: **CLIP Loss minima vs CKA for different encoder pairs on a toy image, caption pair dataset**. We plot the CLIP loss after 500 iterations vs CKA for different image, text encoders and find that a negative correlation exists between CKA and ease of alignment.

**Ease of Alignment with real embeddings:** We investigate whether an inverse relationship exists between the minima of CLIP Loss and CKA when the embeddings are from real data and real encoders. We consider 35 different sentence encoders and 25 different vision encoders and sample 3000 different image,caption pairs from the COCO validation set and pass them through all possible encoder combinations to produce 600 different sets of A and B. We then calculate CKA and compute the minima of CLIP Loss after 500 iterations for these A, B and plot them in Figure 1 with CKA on the x axis and minima of CLIP loss on the y axis in a log scale. We see that for real-world embeddings of images and captions, there is a strong inverse relationship between CKA and the minima of the CLIP loss providing further evidence that encoders with high CKA could have similar similarity structures making them easy to align using simple projections. More toy examples in A.1 .

# 3 FRAMEWORK

Our framework consists of three main components: (1) Encoder Pair Selection, (2) Dataset Curation, and (3) Lightweight Projector Training.

## 3.1 ENCODER PAIR SELECTION

Inspired by Section 2 we use CKA for selecting the most semantically similar encoder pairs for multimodal alignment. We opted for a linear kernel in the CKA computation after observing that the trends in results were largely consistent between linear and RBF kernels, while the linear kernel offers superior computational efficiency. We measure the CKA between encoder spaces by constructing sets of vision embeddings and text embeddings on the COCO validation set of 5000 image, caption pairs. The COCO validation set is chosen as the reference set for its high semantic alignment between the image content and the caption description. We ablate the use of CKA for encoder pair selection in 4.1 and find a positive correlation between CKA and transfer performance to downstream datasets.

## 3.2 DATASET CURATION

By only training the projection layers to align embedding spaces, our approach requires significantly less data compared to training a CLIP model from scratch. To achieve high-quality alignment, it is essential to use a small but well-curated dataset featuring image-caption pairs with a strong semantic correspondence between the images and the text. Additionally, the dataset must encompass a wide range of concepts to facilitate robust zero-shot domain transfer capabilities. With these requirements in mind, our dataset curation process is structured into three key steps:

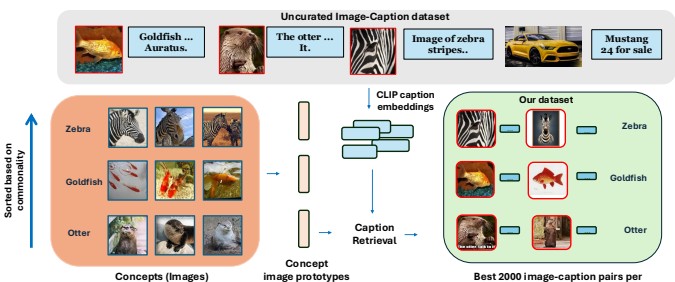

Figure 2: **Overview of our concept-balanced dataset curation process.** Images for each concept are acquired from curated datasets and mapped to CLIP embeddings and averaged to construct Image Prototypes for each concept. Captions of the uncurated dataset are mapped to CLIP's joint embedding space and 2000 samples are picked per concept on the basis of the closest caption embeddings to each concept image prototype.

**Concept Image Prototypes**: Firstly, we collect ∼ 3000 unique concepts from class names of ImageNet, and several other datasets (see A.10.1). Then, we embedded 128 image samples for each concept from the corresponding dataset using CLIP VIT-Large's vision encoder and average them to obtain the concept image prototypes.

**Concept Level Collection**: To create a class-balanced dataset, we first collect image-caption pairs from LAION400M, a large, uncurated source dataset. We then embed all captions using CLIP ViT-Large's text encoder and compute the caption-image prototype similarity for each concept. To ensure diversity, we retrieve 2,000 samples per concept, starting with the less common concepts. As a proxy to establish the commonality of a concept in the pool, we use the average cosine similarity of the top 25,000 captions closest to each concept prototype. This process results in LAION-CLASS-Collected, a high-quality dataset of 6M samples with broad concept coverage. The detailed algorithm is illustrated in Fig 2, and A.5 details the implementation and compute requirements for our collection process.

**Retrieval Datasets**: The LAION-CLASS-Collected dataset offers high concept diversity, but LAION itself is uncurated, with many captions poorly aligned with their images Fan et al. (2024); Nguyen et al. (2024); Chen et al. (2023b). While concept coverage is crucial for strong zero-shot classification, image quality, text diversity, and image-caption alignment are key for effective zero-shot image-text retrieval. In contrast, datasets like CC3M Sharma et al. (2018), CC12M Changpinyo et al. (2021), and SBU Ordonez et al. (2011) feature higher-quality images and better image-caption

alignment than LAION. By combining these, we create a 20M MIX-CLASS-Collected dataset that balances concept coverage with image-text similarity, enhancing both retrieval performance and zero-shot domain transfer across various classification tasks. We examine the impact of each data source on task performance in Sec 4.3.

### 3.3 PROJECTOR ARCHITECTURE

We train lightweight projectors using contrastive loss between adapted image and text embeddings while keeping the unimodal encoders frozen. Figure 3 shows our projector architecture/configuration. We use a lightweight Token Projector Mukhoti et al. (2023) with linear and non-linear branches in a residual configuration for both local tokens and the CLS token of each encoder. The projector's weights are shared for local tokens and separate for the CLS token. Adapted local tokens are averaged and added to the adapted CLS token to form a global embedding, capturing both global and local encoder information.

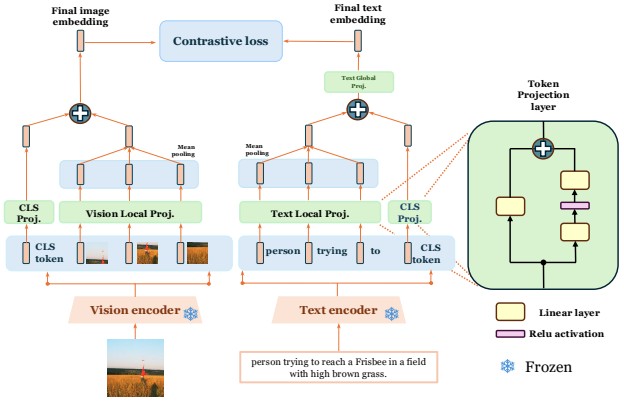

Figure 3: **Lightweight Projector Architecture.** We train only Projection Layers to align modalities. Separate projectors are applied on both the local tokens and the CLS token for each encoder and then combined in a residual manner.

For sentence-transformer architectures, Token Projectors are applied to the tokens, followed by a 2-layer MLP as a global Text Projector, as the text embeddings need further adaptation to become more aligned with the vision embeddings. All projector choices are thoroughly ablated in Section 4.2. Training information and hyperparameters are detailed in A.6.

## 4 ABLATION EXPERIMENTS

We present a set of ablations to validate different components of our pipeline empirically: CKA for encoder selection 4.1, the projector architecture and configuration 4.2, the alignment datasets, and the impact of class-collected data 4.3. We evaluate on downstream tasks like 0-shot domain transfer to Imagenet classification and COCO / Flickr30k image-text retrieval scores.

### 4.1 EFFECTIVENESS OF CKA FOR ENCODER PAIR SELECTION

We train our projector configurations on various combinations of unimodal encoders using the COCO dataset and evaluate image/text retrieval accuracies on the Flickr30k test set, plotting these against CKA scores in Figure 5. The CKA, calculated on the COCO image-caption pairs, shows a strong correlation with retrieval accuracy, indicating that higher semantic similarity, as measured by CKA, predicts better alignment in image/text retrieval. Our findings suggest that CKA can effectively predict which encoder pairs will align well with projector training. The DINOv2-Large and CLIP-ViT-Large-text combination achieves the highest retrieval score, but certain unimodal pairs, like DINOv2-Large and All-Roberta-

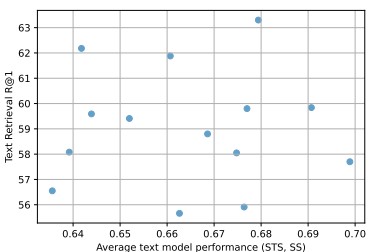

Figure 4: **Unimodal performance does not track alignment performance**

Large-v1 (CKA = 0.69), perform nearly as well. This indicates that these unimodal encoders are highly effective for vision-language alignment, leading us to choose the DINOv2-Large and All-Roberta-Large-v1 pair for larger-scale experiments. Image Retrieval performance is illustrated in A.4.

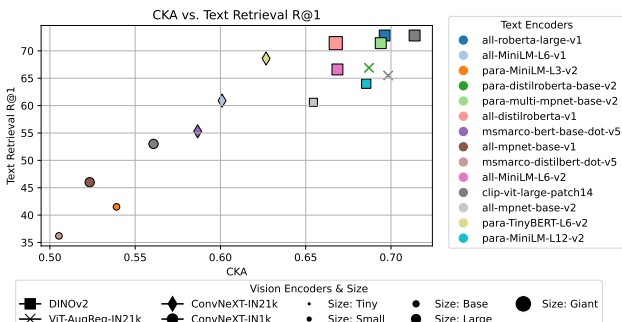

Figure 5: **Retrieval performance vs. CKA for different encoder pairs.** Text retrieval accuracies on Flickr30k are compared to CKA, calculated on the COCO val set. Models are trained on the COCO train set. A clear correlation exists between CKA and alignment quality, as reflected in the retrieval accuracies.

| V Proj. Local | V Proj. CLS | T Proj. Local | T Proj. Global | INet 0-shot |
|---|---|---|---|---|
| mlp | identity | identity | identity | 68.81 |
| token | identity | identity | identity | 68.84 |
| token | identity | identity | mlp | 70.90 |
| token | identity | patch | identity | 71.85 |
| token | identity | token | mlp | 72.15 |
| identity | token | token | mlp | 75.53 |
| token | token | token | mlp | **76.12** |

Table 1: **Projector ablations.**

| Data Source | N | ImageNet | I2T | T2I |
|---|---|---|---|---|
| LAION-CLASS-Collected | 6M | 76.12 | 52.70 | 42.48 |
| CC3M, CC12M, SBU | 14M | 54.17 | 85.30 | 72.44 |
| Both | 20M | 75.04 | 81.32 | 71.38 |
| Both longer training | 20M | **76.30** | **87.54** | **74.17** |

Table 2: **Ablation of Alignment Training Data.**

**Unimodal Performance Does Not Reflect Alignment Quality:** We perform the same ablation as above using DINOv2 and 14 different text encoders from the Sentence Transformers library Reimers & Gurevych (2019). In Fig. 4, we plot Flickr30k text retrieval accuracies against text encoder performance averaged over sentence embedding (STS) tasks (14 datasets) and semantic search (SS) tasks (6 datasets). The results show that text encoder performance does not predict alignment quality, suggesting that CKA, rather than unimodal performance, can be used to identify encoder pairs that easily align. Further ablations are discussed in A.4.

## 4.2 IMPACT OF PROJECTOR ARCHITECTURES

We ablate our projector combinations for the DINOv2 and All-Roberta-Large-v1 encoders by training the projectors to convergence on the LAION-Class-Collected dataset and evaluating the performance on ImageNet 0-shot domain transfer. An MLP applied solely to the local vision tokens achieved 68.81% accuracy, while a Token projection Mukhoti et al. (2023) performed slightly better. Therefore, we used the Token projector for all tokens, both visual and textual. Adding projectors to the text side, targeting both text tokens and a global projector on the averaged local tokens (rows 3, 4, and 5), resulted in performance improvements. These projectors help transform the unimodal text encoder's language-only representations to be more similar to the visual representations. Introducing projectors to the CLS token (row 6) of the visual encoder led to a significant performance increase from 72.15% to 75.13%. Using both CLS and patch projectors in tandem yielded the best performance at 76.12%. This improvement is attributed to DINOv2's dual training objectives: the image-level DINO Caron et al. (2021) objective on the CLS token and the patch-level iBOT Zhou et al. (2021) objective on the patch tokens learning effective global and local features.

## 4.3 IMPACT OF CLASS-COLLECTED DATA / RETRIEVAL DATA

In this section, we ablate the different components of our alignment data. Specifically, we compare the high concept coverage LAION-CLASS-Collected dataset with the higher image-caption quality retrieval datasets: CC3M, CC12M, and SBU. Our experiments show that aligning DINOv2 and All-Roberta-Large-v1 on the high concept coverage dataset results in a high ImageNet zero-shot domain transfer accuracy of 76.1 %, though the retrieval accuracies are lower, at 52.7%/42.2%. In contrast, training with the higher image-caption quality retrieval datasets results in high image and text retrieval scores on the Flickr30k val set (85.3% and 72.4%, respectively). However, the limited concept coverage of these datasets leads to a much lower ImageNet accuracy of 54.1%. Combining both types of datasets yields both high ImageNet accuracy and high image/text retrieval accuracies. To ensure that the extra data is adequately utilized, we train for an additional 30 epochs. This approach results in our best-performing model, achieving an ImageNet accuracy of 76.30% and Flickr retrieval scores of 87.54%/74.17% (last row).

| Model | N | ImageNet | ImageNetv2 | Caltech | Pets | Cars | Flowers | Food | Aircrafts | SUN | CUB | UCF101 |
|---|---|---|---|---|---|---|---|---|---|---|---|---|
| LAION-CLIP VIT-L | 400M | 72.7 | 65.4 | 92.5 | 91.5 | **89.6** | 73.0 | 90.0 | 24.6 | 70.9 | **71.4** | 71.6 |
| OpenAI-CLIP VIT-L | 400M | 75.3 | **69.8** | 92.6 | **93.5** | 77.3 | 78.7 | **92.9** | **36.1** | 67.7 | 61.4 | **75.0** |
| LiT L16L | 112M | 75.7 | 66.6 | 89.1 | 83.3 | 24.3 | 76.3 | 81.1 | 15.2 | 62.5 | 58.7 | 60.0 |
| DINOv2-MpNet (Ours) | 20M | 74.8 | 68.0 | 91.8 | 91.7 | 71.0 | 75.8 | 87.5 | 23.0 | 71.9 | 63.2 | 71.0 |
| DINOv2-ARL(Ours) | 20M | **76.3** | 69.2 | **92.8** | 92.1 | 73.9 | 78.4 | 89.1 | 28.1 | **72.6** | 66.1 | 73.2 |

Table 3: **0-shot domain transfer to classification datasets.** We compare the performance of our DINOv2-ARL projector model, trained on a 20M dataset, against CLIP models from OpenAI and LAION across various datasets. Despite the smaller training size, our model achieves a 76.3% accuracy on ImageNet, outperforming comparably sized CLIP models.

## 5 RESULTS

We evaluate the alignment between vision and text encoders across benchmarks commonly used for CLIP-like models, including zero-shot image classification, image retrieval, localization, multilingual classification/retrieval, and dense caption image-text retrieval. We demonstrate that aligning unimodal vision-language encoders can match or exceed the performance of large CLIP models, despite using smaller datasets and less compute. Additionally, our alignment framework is flexible, enabling the use of specialized encoders for specific tasks, such as aligning multilingual text encoders for multilingual or low-resource image classification/retrieval, or long-context text encoders for dense image/caption retrieval. Furthermore, aligning DINOv2 with a text encoder improves image localization beyond CLIP's vision encoder due to DINOv2's superior localization features.

### 5.1 0-SHOT CLASSIFICATION AND RETRIEVAL

| Model | Flickr | | COCO | |
|---|---|---|---|---|
| | I2T | T2I | I2T | T2I |
| LAION-CLIP VIT-L | **87.6** | 70.2 | 59.7 | 43.0 |
| OpenAI-CLIP VIT-L | 85.2 | 64.9 | 56.3 | 36.5 |
| LiT L16L | 73.0 | 53.4 | 48.5 | 31.2 |
| DINOv2-MpNet (Ours) | 84.6 | 71.2 | 58.0 | 42.6 |
| DINOv2-ARL (Ours) | 87.5 | **74.1** | **60.1** | **45.1** |

Table 4: **Image, Text Retrieval on COCO/Flickr30k.** Our model shows comparable text retrieval scores and significantly better image retrieval results.

| Model | Pascal VOC | Pascal Context |
|---|---|---|
| OpenAI-CLIP-VIT-L* | 23.46 | 14.25 |
| SPARC | 27.36 | 21.65 |
| DINOv2-ARL | **31.37** | **24.61** |

Table 5: **0-shot semantic segmentation mean IOU.** The table shows significant improvements by DINOv2-ARL, even without fine-grained alignment loss. * uses MaskCLIP trick.

Tables 3 and 4 report our model's performance on zero-shot domain transfer to image classification datasets and zero-shot image-text retrieval on the Flickr30k and COCO datasets, respectively. Similar to Maniparambil et al. (2023) we use classwise Visually Descriptive Text (VDT) prompts to enable the unimodal-text encoder in our DINOv2-ARL projector model to better identify the zero-shot classes of the downstream datasets. Detailed descriptions of the evaluation datasets can be found in the A.10, highlighting dataset domains, sizes, and prompt descriptions. We see that despite being trained on a 20M dataset our DINOv2-ARL projector model achieves an ImageNet accuracy of 76.3 % which is 1 % and 3.6 % better than comparably sized CLIP models from OpenAI Radford et al. (2019) and LAION Schuhmann et al. (2021) respectively. Our DINOv2-ARL model demonstrates competitive performance across various datasets compared to LAION and OpenAI CLIP models. The relative performance of these models varies depending on the specific dataset. For example, on the Stanford Cars dataset, LAION-400m Schuhmann et al. (2021) CLIP outperforms OpenAI CLIP by a significant margin of over 12%. Conversely, for the Aircrafts dataset, both OpenAI CLIP and our DINOv2-ARL model show superior performance compared to LAION-400m CLIP. We believe this to be due to the differences in concept coverage for these particular datasets between the LAION400m, OpenAI WIT, and our MIX-CLASS-Collected datasets.

In zero-shot text retrieval, our model slightly outperforms or matches the next best CLIP model, LAION400M-CLIP VIT-L, with scores of 87.5% vs 87.6% on Flickr and 59.7% vs 60.1% on COCO. For image retrieval, our models show a significant advantage, achieving scores of 74.1% vs 70.2% on Flickr and 45.1% vs 43.0% on COCO. This improvement is likely due to the superior quality of the unimodal features produced by the DINOv2 and All-Roberta-Large-v1 encoders, compared to those of the multi-modal vision and text embeddings in the CLIP models.

## 5.2 0-SHOT LOCALIZATION

One key advantage of leveraging frozen unimodal vision and text encoders is the enhancement provided by unimodal features. Specifically, the DINOv2 vision encoder's robust localization capabilities enhance the joint embedding space of the DINOv2-ARL model when trained solely with projectors. We assess this through zero-shot segmentation performance, similar to the Bica et al.; Mukhoti et al. (2023), as shown in Table 5. Our approach involves computing cosine similarities between each patch and all the ground truth classes and subsequently upscaling to the target size. Each patch is then classified into a corresponding class. Consistent with previous studies, the intersection over union (IoU) is computed solely for the foreground classes. In the zero-shot segmentation process of CLIP models, we employ a technique similar to Zhou et al. (2022) to alleviate the opposite visualization problem in CLIP models Li et al. (2023). The patch embeddings from the penultimate layer are passed through the value layer and output MLP of the final self-attention block, followed by projection into the joint embedding space using the vision projector. Meanwhile, our DINOv2-ARL model considers patch embeddings projected into the joint embedding space by the patch projector and augments them with the projected CLS token in a residual manner.

Our DINOv2-ARL model demonstrates superior performance compared to jointly trained dual encoder models like OpenAI's CLIP, achieving over 8% improvement on Pascal VOC and over 10% on Pascal Context. Notably, models utilizing a fine-grained alignment loss like SPARC Bica et al. show improvements over CLIP. However, our DINOv2-ARL model outperforms SPARC by 4% on VOC and 3% on Context datasets. This underscores that the strong localization abilities of DINOv2 patch embeddings are retained even without training with a fine-grained alignment loss. We hypothesize that the localization performance could benefit from the quality of patch embeddings and a more precise localization alignment. Exploring fine-grained losses like SPARC with projector-only CLIP models presents an exciting direction for enhancing localization capabilities in VLMs.

## 5.3 MULTI-LINGUAL RESULTS

| model | | | classification | | | | | | retrieval | | | |
|---|---|---|---|---|---|---|---|---|---|---|---|---|
| | EN | DE | FR | JP | RU | average | EN | DE | FR | JP | RU | average |
| nllb-clip-base@v1 | 25.4 | 23.3 | 23.9 | 21.7 | 23.0 | 23.5 | 47.2 | 43.3 | 45.0 | 37.9 | 40.6 | 42.8 |
| M-CLIP/XLM-Roberta-Large-Vit-B-32 | 46.2 | 43.3 | 43.3 | 31.6 | 38.8 | 40.6 | 48.5 | 46.9 | 46.1 | 35.0 | 43.2 | 43.9 |
| M-CLIP/XLM-Roberta-Large-Vit-L-14 | 54.7 | 51.9 | 51.6 | 37.2 | 47.4 | 48.6 | 56.3 | 52.2 | 51.8 | 41.5 | 48.4 | 50.0 |
| xlm-roberta-base-ViT-B-32@laion5b | 63.0 | 55.8 | 53.8 | 37.3 | 40.3 | 50.0 | 63.2 | 54.5 | 55.7 | 47.1 | 50.3 | 54.2 |
| nllb-clip-large@v1 | 39.1 | 36.2 | 36.0 | 32.0 | 33.9 | 35.4 | 59.9 | 56.5 | 56.0 | **49.3** | 50.4 | 54.4 |
| M-CLIP/XLM-Roberta-Large-Vit-B-16Plus | 48.0 | 46.1 | 45.4 | 32.9 | 40.3 | 42.5 | 63.2 | **61.4** | 59.3 | 48.3 | **54.8** | 57.4 |
| ViT-L-14@laion400m | 72.3 | 48.2 | 49.9 | 2.7 | 4.5 | 35.5 | 64.5 | 26.7 | 38.3 | 1.4 | 1.7 | 26.5 |
| openai/clip-vit-large-patch14 | **75.6** | 46.7 | 49.6 | 6.6 | 3.5 | 36.4 | 59.4 | 19.9 | 28.5 | 4.1 | 1.3 | 22.6 |
| DINOv2-MpNet (Ours) | 73.4 | **61.6** | **58.3** | **43.2** | **49.3** | **57.1** | **70.7** | 60.6 | **60.6** | 45.6 | 52.7 | **58.0** |

Table 6: **Multilingual Classification and Image-Caption Retrieval.** Performance comparison of DINOv2-MpNet with various CLIP models and multilingual baselines on multilingual ImageNet and XTD datasets. Despite being trained only on English data, DINOv2-MpNet outperforms models trained on multiple languages. The upper half of the tables shows multilingual-trained models, while the lower half lists models trained only on English data.

Our framework supports flexible swapping of text encoders, enabling multi-lingual capabilities through multi-lingual encoders, particularly beneficial for low-resource languages. We demonstrate this by aligning DINOv2-Large with paraphrase-multilingual-v2, chosen for its high CKA compatibility, using only English image-caption pairs. We then evaluated our model's performance on multi-lingual image retrieval using the XTD dataset Aggarwal & Kale (2020b) and classification using the ImageNet dataset. For classification, we translated VDT prompts to the languages being considered using the nllb-200-distilled-600M Costa-jussà et al. (2022) model and applied them uniformly across all models.

Multi-lingual classification and retrieval results for five representative languages, are presented in Table 6. Detailed results are in Tables A.4, A.3. The lower section lists models trained exclusively with English captions, specifically the CLIP-VIT-L models trained on the WIT dataset Radford et al. (2021) and the LAION400M dataset Schuhmann et al. (2021). The upper sections feature models trained with translated captions, such as CLIP models based on LAION5B Schuhmann et al. (2022), M-CLIP models Chen et al. (2023a), and NLLB-CLIP models Visheratin (2023).

Our DINOv2-MpNet, trained solely on English image-caption pairs, outperforms other English-only CLIP models by over 31% in average retrieval performance across five languages and by 6% in English. While English CLIP models perform well on Latin script languages, their performance drops for non-Latin languages like RU and JP due to the English-only tokenizer. In contrast, our DINOv2-MpNet remains competitive across both Latin and non-Latin languages, even against models trained on multilingual data. Notably, it outperforms the laion5b-trained xlm-roberta-base-VitB32 by 0.6%, despite using only 20 million English image-caption pairs compared to the 5B multilingual pairs in LAION5B. In classification tasks, DINOv2-MpNet surpasses the LAION400m-trained ViT-L on English Imagenet, delivering significantly better results (over 20% on average) across five languages. Among multilingual models, it exceeds both nllb-clip and M-CLIP models, surpassing the next best M-CLIP/XLM-Roberta-Large-Vit-L-14 by over 8%, despite not using any multilingual text data. It also outperforms the LAION5B-trained CLIP model by 7% despite its use of multilingual image-caption pairs. This underscores the efficiency of our training approach, achieving highly performant models with significantly fewer image-caption pairs, and suggests that further training on translated pairs could enhance DINOv2-MpNet's performance, particularly in low-resource languages.

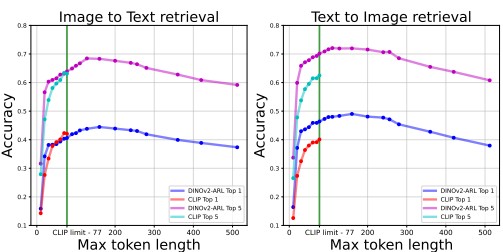

| Model | Data | SS | Trainable / Total | Compute | IN 0-shot |
|---|---|---|---|---|---|
| OpenAI CLIP | 400M | 12.8B | 427M / 427M | 21,845 | 72.7% |
| LAION400M CLIP | 400M | 12.8B | 427M / 427M | 25,400 | 75.3% |
| DINOv2-ARL | 20M | 0.6B | 11.5M / 670M | 400 | 76.3% |

Table 7: **Compute requirements, Dataset size, and Number of trainable parameters are orders of magnitude lower when using projectors to align semantically similar encoders.** By using projectors to align semantically similar encoders, compute requirements drop 65-fold, dataset size shrinks by 20 times, and only 1% of total parameters are trainable while outperforming other CLIP models. Compute measured in GPU hours on an A100 (80 GB) GPU.

Figure 6: Retrieval performance comparison between DINOv2-ARL encoder pair and OpenAI CLIP as the maximum token length increases. The vertical green line indicates the standard CLIP token limit of 77.

## 5.4 DENSELY CAPTIONED IMAGES (DCI) DATASET AND LONG-TEXT RETRIEVAL

The Densely Captioned Images (DCI) dataset Urbanek et al. (2024) offers a unique approach to image-text datasets, featuring 7,805 natural images with richly annotated, mask-aligned descriptions averaging over 1,000 words per image. This level of detail provides an opportunity to explore the limits of vision-language models in processing long-term textual information in relation to visual content. While DCI includes its own benchmarks using summarized captions, our focus is on image-text and text-image retrieval tasks using the entire dataset without summarization or subcropping, allowing us to investigate the long-text retrieval capabilities of our framework.

To demonstrate the advantages of processing longer captions, we conducted an experiment varying the maximum token length allowed by the tokenizer. As shown in Figure 6, our DINOv2-ARL encoder pair achieves comparable performance to OpenAI CLIP at the standard limit of 77 tokens. However, our approach's strength becomes evident as we extend beyond this limit, with consistent improvement in retrieval accuracy up to approximately 200-300 tokens. These results highlight our framework's ability to effectively utilize longer, more detailed captions for improved retrieval, capturing nuanced details and context that may be lost when constrained to shorter text sequences.

## 5.5 TRAINING COMPUTE

We report the Alignment Training compute requirements for different models in 7. We see that aligning pre-trained vision, language encoders to get a competitive CLIP like model requires only 50 hours of training with 8 A100 GPUS which is almost a 65 fold reduction in the amount of training compute. This makes the development of multi-modal models accessible to the wider research community as well as reducing the environmental impact of training highly performant multi-modal models by reusing strong publicly available uni-modal models. Since we only need to train 11.5M of the total 670M parameters (about 1 %) we can train with a much smaller and denser dataset reducing the data requirements to 20M which is 20 fold decrease in dataset requirement compared to CLIP models from LAION and OpenAI making our framework useful for training performant

multi-modal models in various domains like mutli-modal systems for low-resource languages, 3D model search systems, fMRI to Image model mapping systems and many more. Despite the reduced compute and data requirements for alignment our model outperforms both CLIP models compared on domain transfer to Imagenet as well as image, text retrieval.

# 6 RELATED WORKS

**Multimodal Pretraining:** The CLIP models from OpenAI Radford et al. (2021) and ALIGN Jia et al. (2021) pioneered using web-scale image-caption data to align image and text modalities via an InfoNCE Oord et al. (2018) loss, optimizing mutual information between embeddings. LAION Schuhmann et al. (2021; 2022) replicated this approach in the open domain, open-sourcing pre-training datasets. While these models excel in zero-shot tasks, they demand substantial computational resources, around 20k GPU hours. Taking advantage of the recent improvements in the representation quality of unimodal encoders such as DINOv2 Oquab et al. (2023) (vision) and Sentence Transformer Reimers & Gurevych (2019) (language) models, Zhai et al. (2022) reduce the training cost by locking the image encoder and training only the text encoder to achieve competitive performance. Similarly, Khan & Fu (2023) further aligned frozen uni-modal encoders using projection layers, BitFit Zaken et al. (2021), and trainable adapters, but their approach is sub-optimal compared to CLIP, likely due to smaller datasets used and random encoder pair selection. In contrast, in this work, we strive to identify the best encoder pairs for alignment first and then scale up projector-only training to improve the multimodal alignment.

**Representational Similarity:** Recent studies show that the semantic similarity between vision and language model embeddings is high for several model pairs. Maniparambil et al. (2024) reports that this similarity, measured by Centered Kernel Alignment Kornblith et al. (2019), increases with more training data for vision models. Similarly, Huh et al. (2024) finds that better-performing language models have higher semantic similarity to the DINOv2 Oquab et al. (2023) vision model. These similarities have been leveraged for 0-shot and multi-lingual retrieval tasks using strong uni-modal encoders without additional training Maniparambil et al. (2024); Moschella et al. (2022), though scalability is an issue. Additionally, Merullo et al. (2022) demonstrates that a simple linear mapping allows a frozen language model to interpret visual input, provided the visual encoder aligns with language concepts (e.g., CLIP). Similarly, Dwivedi & Roig (2019); Dwivedi et al. (2020) also uses representational similarity metrics to identify pre-trained models for effective transfer to downstream tasks. These findings suggest that a simple projection transformation separates the embedding spaces of well-trained vision and language models, motivating our work on developing CLIP models using projection layers between semantically similar encoder pairs.

**Automatic Data Curation:** Our dataset curation pipeline draws on various approaches in Vision-Language dataset construction Radford et al. (2021); Gadre et al. (2024); Xu et al. (2024). Radford et al. (2021) used image metadata to gather high-quality image-caption pairs, while Schuhmann et al. (2021) replicated the CLIP dataset by filtering with pretrained vision encoders. Recent methods like Gadre et al. (2024) employ CLIP-based filtering and ad hoc filtering techniques, and Xu et al. (2024) mimics CLIP's data collection via metadata retrieval. Similarly, Oquab et al. (2023) uses a pretrained vision encoder to curate web images most similar to images in curated datasets. Our approach is similar, constructing concept image prototypes from few-shot labeled examples and retrieving relevant web images from the LAION-400M pool using CLIP caption embeddings, avoiding the computational cost of generating vision embeddings for the entire dataset.

# 7 CONCLUSION

Our research introduces a paradigm shift in vision-language alignment, demonstrating that state-of-the-art performance can be achieved with a fraction of the resources traditionally required. By leveraging the latent compatibility of well-trained unimodal encoders, we have unlocked a new direction in efficient multimodal AI development.

Future work in this area could explore fine-grained alignment techniques, optimize projection architectures, and expand to other modalities beyond vision and language. By democratizing multimodal AI research, our framework has the potential to accelerate innovation and reshape approaches to multimodal AI development.

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

# A APPENDIX

## A.1 TOY EXAMPLE USING RANDOM LATENT MODEL

Similar to Sec. 2.2 here we investigate whether semantically similar encoder embedding spaces can be aligned through a simple projection transformation, using a random latent model.

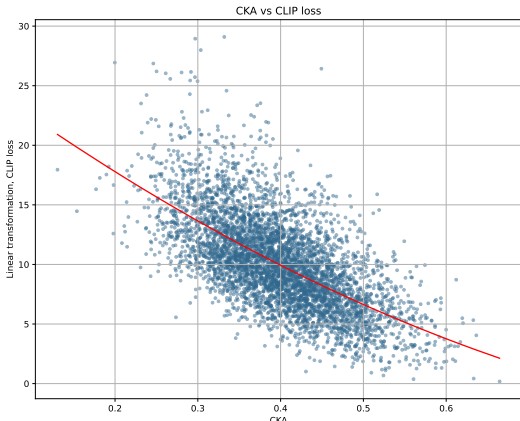

Figure A.1: **CLIP Loss minima are negatively correlated to CKA.** We plot CKA vs CLIP Loss for random instances of A and B.

```
# Init Z with random values scaled to
    ↪ [-1, 1]
Z = 2 * rand(n, d) - 1

# Define non-linear transforms T1 and
    ↪ T2
T1, T2 = NLTransform(d, d),
    ↪ NLTransform(d, d)

# Sample random weights w1 and w2
w1, w2 = rand(1), rand(1)

# Compute A and B using transforms
A = T1(Z) + w1 * rand(n, d)
B = T2(Z) + w2 * rand(n, d)
```

Figure A.2: **Code for initializing A and B from a latent world model Z.** Random instances of A, B are generated using random non-linear transformations of latent vector Z denoting a representation of the real world.

In our experiment, we generated $10^3$ instances of two vector sets, $A$ and $B$, each containing 32 vectors of 16 dimensions. Following the approach in Maniparambil et al. (2024); Huh et al. (2024), we modeled the world using a latent distribution $Z$, with Image and Text representations ($A$ and $B$) as random independent non-linear transformations from $Z$ with additive noise. For each sampled pair of $A$ and $B$ matrices, we calculated the CKA and the minimum CLIP loss. The non-linear transform was defined as a randomly initialized 2-layer MLP with ReLU non-linearity and hidden dimensions significantly larger than the input dimensions, ensuring it could universally approximate the non-linear transformation Hornik et al. (1989). Figure A.2 was used to generate each instance.

Figure A.1 illustrates the results of this experiment, showing a clear negative correlation between CKA and minima of the CLIP loss. As CKA increases, indicating greater similarity between the similarity structures of A and B, the minima of CLIP loss consistently decreases. Despite arising from a simplified experiment, the observed strong inverse relationship between CKA and CLIP loss provides empirical support for using CKA as a predictor of alignment potential between embedding spaces. Since CLIP loss is lower-bounded by mutual information, and mutual information is correlated with HSIC, higher CKA suggests a stronger alignment between embeddings. This implies that the achievable minima of CLIP loss is lower when the embedding spaces already have a higher CKA, reflecting greater mutual information and ease of alignment.

## A.2 CKA VS GRAPH STRUCTURE

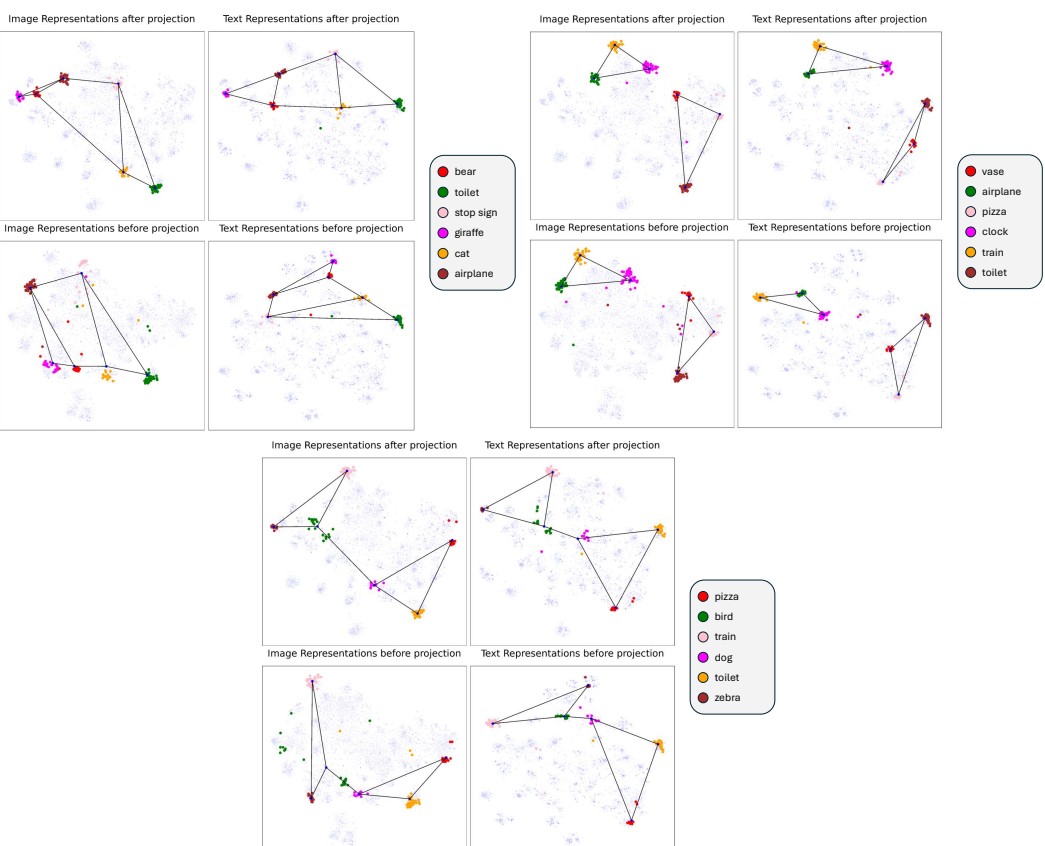

Figure A.3: TSNE visualizations of encoder outputs for six COCO detection classes. Left: DINOv2 (vision), Right: All-Roberta-Large-v1 (text).

To visually demonstrate how CKA represents similarities in graph structures across different encoder spaces, we conducted an experiment using the MSCOCO validation set. We examined encoder outputs for DINOv2 and All-Roberta-Large-v1, before and after projection, focusing on relationships between formed clusters in both domains. For each cluster, we identify COCO detection class and COCO image-caption pairs where the image contained only the respective class among its detection annotations. We then extracted encoder outputs for these samples from both vision and text encoders, before and after applying our projection layers, and applied the TSNE algorithm to visualize their structure in a lower-dimensional space. For each visualization, we pick 6 classes to highlight the shape similarities between graphs of encoder spaces.

Figure A.3 shows the resulting TSNE visualizations for the six selected classes across four conditions: vision pre-projection, vision post-projection, text pre-projection, and text post-projection. The visualizations reveal striking similarities in cluster shapes and relative positions across the different encoder spaces, particularly before projection. This visual similarity aligns with our quantitative CKA results, providing an intuitive illustration of how CKA captures structural similarities between different embedding spaces.

## A.3 COMPARISON TO LILT

Tables A.1 and A.2 report the zero-shot domain classification and retrieval performance of LiLT models Khan & Fu (2023). The vision encoder is initialized with the DeiT base model Touvron et al. (2021), and the text encoder is from SimCSE Gao et al. (2021). The LilT$_{DA}$-base model is trained by duplicating and appending the last transformer layer, while only unlocking the last encoder and projector layers. The LilT$_{LwA}$-base model introduces trainable layerwise adapters for

| Model | N | ImageNet | ImageNetv2 | Caltech | Pets | Cars | Flowers | Food | Aircrafts | SUN | CUB | UCF101 |
|---|---|---|---|---|---|---|---|---|---|---|---|---|
| LAION-CLIP VIT-L | 400M | 72.7 | 65.4 | 92.5 | 91.5 | **89.6** | 73.0 | _90.0_ | 24.6 | 70.9 | **71.4** | 71.6 |
| OpenAI-CLIP VIT-L | 400M | 75.3 | **69.8** | _92.6_ | **93.5** | _77.3_ | 78.7 | **92.9** | **36.1** | 67.7 | 61.4 | **75.0** |
| LiT L16L | 112M | _75.7_ | 66.6 | 89.1 | 83.3 | 24.3 | 76.3 | 81.1 | 15.2 | 62.5 | 58.7 | 60.0 |
| LilT$_{DA}$-base | 0.5M | 15.9 | 12.9 | 37.6 | 7.2 | 1.6 | 1.1 | 13.3 | 1.7 | 25.6 | 2.3 | 19.1 |
| LilT$_{LwA}$-base | 0.5M | 14.4 | 12.1 | 42.3 | 4.4 | 1.3 | 2.1 | 12.3 | 1.6 | 26.5 | 1.4 | 26.6 |
| DINOv2-MpNet (Ours) | 20M | 74.8 | 68.0 | 91.8 | 91.7 | 71.0 | 75.8 | 87.5 | 23.0 | _71.9_ | 63.2 | 71.0 |
| DINOv2-ARL(Ours) | 20M | **76.3** | _69.2_ | **92.8** | _92.1_ | 73.9 | _78.4_ | 89.1 | _28.1_ | **72.6** | _66.1_ | _73.2_ |

Table A.1: **0-shot domain transfer to classification datasets.** We compare the performance of our DINOv2-ARL projector model, trained on a 20M dataset, against CLIP models from OpenAI and LAION across various datasets. Despite the smaller training size, our model achieves a 76.3% accuracy on ImageNet, outperforming comparably sized CLIP models.

both the vision and text encoders. LiLT public checkpoints are trained on 500k image-caption pairs from the COCO dataset. However, LiLT's performance lags behind CLIP models and our DINOv2-ARL projector model, primarily due to suboptimal encoder pairs and limited concept coverage in the COCO training set for alignment.

## A.4 ENCODER PAIRS ABLATIONS

Similar to Sec 4.1, we train our projector configurations on various combinations of unimodal encoders using the COCO dataset and evaluate image/text retrieval accuracies on the Flickr30k test set, plotting these against CKA scores. In Fig. A.4 both the Image and Text retrieval accuracies shows a strong correlation with CKA suggesting that CKA can effectively predict which encoder pairs will align well with projector training.

A naive approach to choosing the best encoder pair is to chose the unimodal encoders with highest performance in their respective modalities, but it's not straightforward which benchmarks can be more predictive of ease of

| Model | Flickr | | COCO | |
|---|---|---|---|---|
| | I2T | T2I | I2T | T2I |
| LAION-CLIP VIT-L | **87.6** | 70.2 | 59.7 | 43.0 |
| OpenAI-CLIP VIT-L | 85.2 | 64.9 | 56.3 | 36.5 |
| LiT L16L | 73.0 | 53.4 | 48.5 | 31.2 |
| LilT$_{DA}$-base | 47.6 | 34.46 | 41.4 | 29.1 |
| LilT$_{LwA}$-base | 56.8 | 41.7 | 47.0 | 33.7 |
| DINOv2-MpNet (Ours) | 84.6 | 71.2 | 58.0 | 42.6 |
| DINOv2-ARL (Ours) | 87.5 | **74.1** | **60.1** | **45.1** |

Table A.2: **Image, Text Retrieval on COCO/Flickr30k.** Our model shows comparable text retrieval scores and significantly better image retrieval results.

alignment. To demonstrate this, we consider the same ablation as above, but with DINOv2 and 14 different text encoders from the SentenceTransformers Reimers & Gurevych (2019) library. We consider 2 types of text model benchmarks. 1. Sentence Embedding task or Semantic Textual Similarity (STS) is the task of evaluating how similar two texts are in terms of meaning. These models take a source sentence and a list of sentences and return a list of similarity scores. The task is evaluated using Spearman's Rank Correlation. We average over 14 datasets reported in Reimers & Gurevych (2019; 2024). 2. Semantic Search (SS) is the task of retrieving relevant documents or passages based on the semantic content of a query. Rather than relying solely on keyword matching, semantic search models generate embeddings for both the query and the documents, allowing for retrieval based on contextual and conceptual similarity and is evaluated using Normalized Discounted Cumulative Gain (nDCG), which measure the relevance of retrieved documents in ranked lists. We average over 6 datasets reported in Reimers & Gurevych (2019; 2024).

In Fig A.5, we see that there is a clear correlation (pearson corr.=0.81, p=4e-4) between downstream Flickr30k performance and CKA on the COCO val set, suggesting that CKA is a better predictor of ease of alignment. The average unimodal performance (pearson corr.=0.47, p=0.08), as well as the semantic search (SS) performance (pearson corr.=0.13, p=0.65), are not predictive of the ease of alignment. Meanwhile, Sentence Task Similarity (STS) tasks are more predictive of downstream alignment (pearson corr.=0.72, p=0.003) but still worse than CKA and it's not intuitive which unimodal performance is to be considered.

## A.5 DATA CURATION IMPLEMENTATION DETAILS

We streamline our class collection process by precomputing CLIP text embeddings for LAION-400M and CLIP image prototype embeddings for various concepts, allowing us to run different collection methods without needing to recompute embeddings. The embedding process takes just 12 hours on two nodes with 4 A6000 GPUs each. Class-level collection is performed using GPU-accelerated PyTorch code on a single GPU, completing in under an hour. While image-to-image-

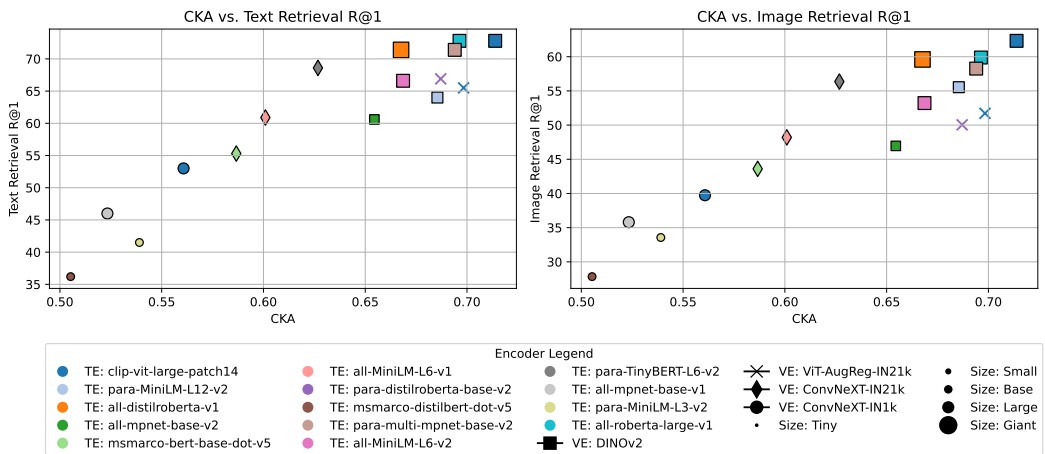

Figure A.4: **Retrieval performance vs. CKA for different encoder pairs.** Text/Image retrieval accuracies on Flickr30k are compared to CKA, calculated on the COCO val set. Models trained on COCO train set. A clear correlation exists between CKA and alignment quality (Pearson correlation = 0.92, p = 2.1e-7), as reflected in retrieval accuracies.

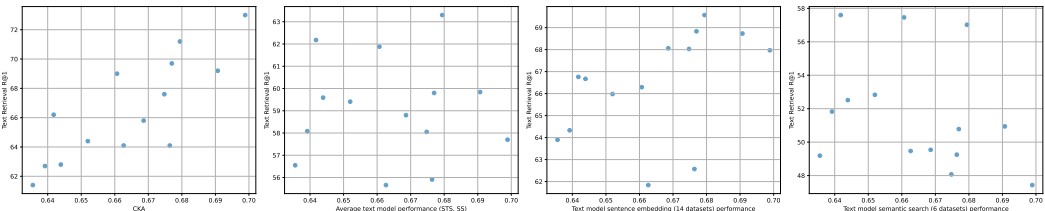

Figure A.5: **Retrieval performance vs. text model performance for DINOv2 and different text encoders.** Text/Image retrieval accuracies on Flickr30k are compared different text encoder tasks performance. CKA is more closely correlated with retrieval performance than text encoder downstream task performance on sentence embedding tasks, semantic search tasks. Models trained on COCO train set.

prototype collection, as in Oquab et al. (2023), could yield higher-quality results, it demands significantly more GPU resources due to the need to create CLIP embeddings for all LAION-400M images. We find that caption-image-concept similarity performs well for image classification accuracy. To support efficient multi-modal model training, we release the LAION-CLASS-Collected parquets for research use.

## A.6    PROJECTOR TRAINING DETAILS

We use the standard CLIP loss with a learnable temperature parameter to train the projectors while keeping the vision and text encoders frozen. For our largest experiments on the 20M MIX-CLASS-Collected dataset, we use an effective batch size of 16k and train for 30 epochs. Training is done with a cosine learning rate scheduler, ramping up to 1e-3 in the first epoch. Additional hyperparameters are detailed in the table in the appendix. The training process takes 50 hours on a node with 8 A100 GPUs.

## A.7    MULTI-LINGUAL FULL RESULTS

Another significant advantage of using only Projectors to align modalities is the ability to swap the text encoder with multi-lingual encoders trained on various languages, thus potentially extending a CLIP model to accommodate any language. This feature is particularly beneficial for low-resource languages. We demonstrate the feasibility of this approach by training projectors to align the DINOv2 visual encoder with the paraphrase-multilingual-v2 text encoder, using a dataset consisting

solely of English image-caption pairs. We selected this specific text encoder as it showed the highest compatibility in terms of CKA with DINOv2. Subsequently, we evaluated the performance of our model on multi-lingual image retrieval using the XTD dataset Aggarwal & Kale (2020a) and on multi-lingual image classification using the ImageNet dataset. For multi-lingual classification, we translate our VDT prompts Maniparambil et al. (2023) to the languages being considered using the nllb-700M model Costa-jussà et al. (2022) and then use the same prompts for all the models being considered including ours.

For both multi-lingual classification and retrieval tasks, our comparisons are structured into two categories as delineated in Table A.4 and Table A.3. The lower sections of each of these tables list models trained exclusively with English captions, more specifically the CLIP-VIT-L models from OpenAI and LAION trained on 400 million image caption pairs of WIT dataset and LAION400M dataset respectively. The upper sections of these tables feature models trained with translated captions, including those employing contrastive training with multi-lingual image-caption pairs such as CLIP-models based on the LAION5B multi-lingual dataset, which contains image-caption pairs in over 100 languages. We also compare against, M-CLIP Chen et al. (2023a) models that are trained using English and translated captions to align a multi-lingual text encoder with CLIP's original text encoder through contrastive learning, thereby enhancing performance on multi-lingual tasks. Additionally we also compare against the NLLB-CLIP Visheratin (2023) models developed through LiT Zhai et al. (2022) techniques, coupling a frozen CLIP visual encoder with an unfrozen multi-lingual text encoder using translated captions from the smaller LAION-COCO dataset. We compare against only model sizes of up to ViT-Large for fair comparison.

**Retrieval results**: Our model DINOv2-MpNet trained only on English image,caption pairs outperforms all other CLIP models trained only on English image caption pairs, by a large margin of over 43 % on average retrieval performance over 10 languages. We also outperform the next best performing English CLIP model trained on LAION400m English caption retrieval by over 6 percent. On Latin script languages the CLIP models have decent performance while it falls significantly for non Latin languages like JP, KO, PL, RU, TR, and ZH. This is mainly because these models were trained using an English only tokenizer which results in unknown token for most characters of these languages. However our DINOv2-MpNet projector model maintains competitive performance on all languages both Latin script and non Latin script even when compared against models specifically trained using multi-lingual data (Upper half of the table). Amongst the multi-lingual trained CLIP models we perform better than laion5b trained xlm-roberta-base-VitB32 by 4.5 percent. It is to be noted here that we only use 20 million Image caption pairs for alignment while LAION5B has over 5B image-caption pairs from over 100 languages and multi-lingual webli has over 30B image-caption pairs from over 100 languages. It is to be noted that our DINOv2-Mpnet is also competitive with M-CLIP model XLM-Roberta-Large-Vit-B-16Plus(56.1 vs 57.7) which has been trained using translated English sentences of over 175 million data points to over 100 languages, and 3M translated image, caption pairs from CC3m.

**Classification results**: We see a similar trend when we compare our DINOv2-MpNet projector model against CLIP baselines(lower section), and multi-lingual baselines (upper section) on multi-lingual imagenet classification in Table. Our model showcases competitive performance to that of OpenAI-clip model while beating LAION400m trained ViT-Large on english Imagenet, while performing significantly better on all other languages considered (over 24 percent better on 8 language average). When compared with models trained with multi-lingual data, our model outperforms both nllb-clip models as well as M-CLIP models, beating the next best performing model M-CLIP/XLM-Roberta-Large-Vit-L-14 by over 3 percent despite not training using any multi-lingual text data. We believe that training using translated image-caption pairs of our dataset would further improve the performance of our method, and we leave this as a future work. The main advantage of training using our methods is that we can get highly porformant CLIP-like models using much lesser amount of image-caption pairs, (more than 20x lesser) resulting in quick adaptation to low resource languages given that a multi-lingual text encoder exists for that language.

| model | EN | DE | ES | FR | IT | JP | KO | PL | RU | TR | ZH | average |
|---|---|---|---|---|---|---|---|---|---|---|---|---|
| nllb-clip-base@v1 | 47.2 | 43.3 | 44.1 | 45.0 | 44.7 | 37.9 | 39.4 | 45.5 | 40.6 | 41.2 | 41.1 | 42.3 |
| M-CLIP/XLM-Roberta-Large-Vit-B-32 | 48.5 | 46.9 | 46.4 | 46.1 | 45.8 | 35.0 | 36.9 | 48.0 | 43.2 | 45.7 | 45.4 | 43.9 |
| M-CLIP/XLM-Roberta-Large-Vit-L-14 | 56.3 | 52.2 | 52.7 | 51.8 | 53.6 | 41.5 | 42.5 | 54.1 | 48.4 | 52.7 | 53.5 | 50.3 |
| xlm-roberta-base-ViT-B-32@laion5b | 63.2 | 54.5 | 54.6 | 55.7 | 55.7 | 47.1 | 43.8 | 55.5 | 50.3 | 48.2 | 50.8 | 51.6 |
| nllb-clip-large@v1 | 59.9 | 56.5 | 56.7 | 56.0 | 55.5 | **49.3** | **51.7** | 57.4 | 50.4 | 56.0 | 52.3 | 54.2 |
| M-CLIP/XLM-Roberta-Large-Vit-B-16Plus | 63.2 | **61.4** | **59.8** | 59.3 | **61.0** | 48.3 | 49.8 | 64.0 | 54.8 | **59.6** | **58.8** | **57.7** |
| ViT-L-14@laion400m_e31 | 64.5 | 26.7 | 31.4 | 38.3 | 26.6 | 1.4 | 0.4 | 4.8 | 1.7 | 4.1 | 1.0 | 13.6 |
| openai/clip-vit-large-patch14 | 59.4 | 19.9 | 26.6 | 28.5 | 19.2 | 4.1 | 0.3 | 3.9 | 1.3 | 2.6 | 0.7 | 10.7 |
| DINOv2-MpNet (Ours) | **70.7** | 60.6 | 59.0 | **60.6** | 60.7 | 45.6 | 49.8 | 58.3 | 52.7 | 55.8 | 57.9 | 56.1 |

Table A.3: **Multilingual image-caption retrieval** performance on XTD dataset. DINOv2-MpNet outperforms many baselines despite English-only training. Upper: multilingual-trained models; Lower: English-only trained models.

| model | EN | AR | ES | FR | DE | JP | ZH | RU | average |
|---|---|---|---|---|---|---|---|---|---|
| nllb-clip-base@v1 | 25.4 | 20.4 | 23.9 | 23.9 | 23.3 | 21.7 | 20.3 | 23.0 | 22.4 |
| nllb-clip-large@v1 | 39.1 | 30.1 | 36.5 | 36.0 | 36.2 | 32.0 | 29.0 | 33.9 | 33.4 |
| M-CLIP/XLM-Roberta-Large-Vit-B-32 | 46.2 | 33.4 | 43.7 | 43.3 | 43.3 | 31.6 | 29.1 | 38.8 | 37.6 |
| M-CLIP/XLM-Roberta-Large-Vit-B-16Plus | 48.0 | 35.1 | 46.6 | 45.4 | 46.1 | 32.9 | 31.3 | 40.3 | 39.7 |
| xlm-roberta-base-ViT-B-32@laion5b | 63.0 | 29.0 | 53.4 | 53.8 | 55.8 | 37.3 | 26.8 | 40.3 | 42.3 |
| M-CLIP/XLM-Roberta-Large-Vit-L-14 | 54.7 | **40.0** | 51.9 | 51.6 | 51.9 | 37.2 | **35.2** | 47.4 | 45.0 |
| ViT-L-14@laion400m_e32 | 72.3 | 6.4 | 44.7 | 49.9 | 48.2 | 2.7 | 2.3 | 4.5 | 22.7 |
| openai/clip-vit-large-patch14 | **75.6** | 6.7 | 46.2 | 49.6 | 46.7 | 6.6 | 2.2 | 3.5 | 23.1 |
| DINOv2-MpNet (Ours) | 73.4 | 38.0 | 56.8 | 58.3 | 61.6 | **43.2** | 33.3 | 49.3 | 48.6 |

Table A.4: **Multi-lingual classification.** Classification performance comparison of DINOv2-MpNet and various CLIP models and multilingual baselines on multilingual ImageNet. Our DINOv2-MpNet model trained only on English data outperforms even models trained on multi-lingual data. The upper half of the table lists models trained on multiple languages, while the lower half lists models trained only on English data. The models are evaluated on translations of the labels and the prompts made using nllb-200-distilled-600M translation model. Costa-jussà et al. (2022)

## A.8 DATASET SCALE

Figure A.6 illustrates that while performance scales with an increasing number of randomly sampled data points from the LAION400M dataset, the rate of improvement diminishes, highlighting the critical need for densely covered and high-quality datasets when training projectors to align modalities. Additionally, the comparative performance of MIX-CLASS-Collected data reveals that datasets curated with more focused criteria can lead to better performance gains than simply increasing the volume of data. This underscores the importance of prioritizing dataset quality over quantity, especially given the observed diminishing returns when using larger data sizes for projector-based alignment.

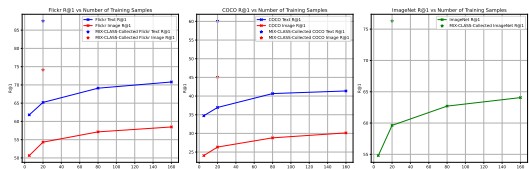

Figure A.6: **Performance scales with higher amounts of randomly sampled LAION data** The performance scales with higher amounts of randomly sample data from LAION400M, but very slowly, highlighting the need for a densely covered and high quality dataset when training projectors only to align modalities.

## A.9 sDCI BENCHMARK RESULTS

We evaluate our method on the Densely Captioned Images (DCI) dataset Urbanek et al. (2024), which contains 7,805 images with mask-aligned descriptions averaging over 1,000 words each. To accommodate current models' token limits, the authors also provide sDCI, a summarized version with CLIP-compatible 77-token captions generated by LLMs.

sDCI introduces several benchmarks:

- All SCM (Subcrop-Caption Matching): Matches captions to corresponding image sub-crops.

| Model | All SCM | All Neg | All Pick5-SCM | All Pick5-Neg | Base Neg | All Hard-Negs |
|---|---|---|---|---|---|---|
| CLIP Baseline | **40.06**% | 60.79% | **11.21**% | **24.06**% | 67.56% | 41.34% |
| DINOv2-ARL (Ours) | 29.33% | **64.36**% | 9.35% | 21.39% | **81.94**% | **61.10**% |

Table A.5: Performance comparison on DCI dataset benchmarks

- All Neg: Distinguishes between positive captions and LLM-generated negatives.

- All Pick5-SCM: Similar to All SCM, but uses multiple captions per subcrop.

- All Pick5-Neg: Distinguishes between multiple positive captions and a negative.

- Base Neg: Focuses on caption-negative distinction for full images only.

- All Hard-Negs: Uses the most challenging LLM-generated negatives.

We tested our DINOv2-ARL model on the sDCI dataset benchmarks. Table A.5 presents our results alongside the CLip baseline. Our method demonstrates competitive performance compared to the CLIP baseline across several DCI benchmarks.

In the Subcrop-Caption Matching tasks (All SCM and All Pick5-SCM), our model performs slightly below the CLIP baseline. This suggests that there is room for improvement in our approach when it comes to distinguishing between the different parts that compose an image.

However, our model shows notable improvements in the negative detection tasks. We outperform CLIP on All Neg (64.36% vs. 60.79%), Base Neg (81.94% vs. 67.56%), and All Hard-Negs (61.10% vs. 41.34%). These results demonstrate the potential of our method in aligning vision and language models for a fine-grained understanding of image content, especially in scenarios requiring robust discrimination between relevant and irrelevant captions. Future work could focus on improving the model's performance on sub-crop caption matching tasks while maintaining its strong capabilities in negative detection.

### A.10   0-Shot Classification and Retrieval Evaluation Datasets

To evaluate the performance of our DINOv2-ARL projector model and compare it with baseline CLIP models, we utilized a diverse set of datasets for zero-shot classification and retrieval tasks. These datasets span various domains and challenge the models' ability to generalize across different visual concepts.

For zero-shot classification, we employed the following datasets:

- ImageNet Deng et al. (2009): A large-scale dataset with 1000 object categories, widely used as a benchmark for image classification tasks. It contains over 1.2 million training images and 50,000 validation images, with each image labeled with one of 1000 object classes.

- ImageNetV2 Recht et al. (2019): A newer version of ImageNet designed to test the robustness of models trained on the original ImageNet. It features 10,000 new test images collected using the same procedure as the original, but addressing certain biases in the original dataset.

- Caltech101 Li et al. (2022): A dataset containing pictures of objects belonging to 101 categories, plus a background category. It includes about 40 to 800 images per category, with most categories having about 50 images. The dataset is known for its high intra-class variability.

- Oxford-IIIT Pet Parkhi et al. (2012): A 37-category pet dataset with roughly 200 images for each class, featuring different breeds of cats and dogs. It includes pixel-level trimap segmentations and breed-level labels for each image.

- Stanford Cars Krause et al. (2013): A dataset of 196 car classes, totaling 16,185 images. Classes are at the level of Make, Model, Year (e.g., 2012 Tesla Model S). It includes 8,144 training images and 8,041 testing images, with bounding box annotations.

- Oxford Flowers102 Nilsback & Zisserman (2008): A 102 category dataset consisting of 102 flower categories common to the UK. It contains 40 to 258 images per class and provides segmentation data for each image. The dataset is particularly challenging due to the fine-grained nature of the categories.

- Food101 Bossard et al. (2014): A large dataset of 101 food categories, with 101,000 images. It features 1000 images per food class, with 250 test images and 750 training images per class. The training images are not manually cleaned, adding a level of noise to the dataset.

- FGVC Aircraft Maji et al. (2013): A fine-grained visual classification dataset with 10,200 images of aircraft, spanning 100 aircraft models. Each model is associated with a specific variant, manufacturer, family, and collection. The dataset includes 6,667 training images and 3,333 test images.

- SUN397 Rouach et al. (2020): A scene recognition dataset with 397 categories and 108,754 images, covering a large variety of environmental scenes under various lighting conditions. It provides at least 100 images per class and has been used extensively for scene recognition tasks.

- Caltech-UCSD Birds-200-2011 (CUB) Wah et al. (2011): A dataset for fine-grained image classification with 200 bird species, containing 11,788 images. Each image has detailed annotations including 15 part locations, 312 binary attributes, and 1 bounding box. It's widely used for fine-grained visual categorization research.

- UCF101 Soomro et al. (2012): An action recognition dataset with 101 action categories, consisting of realistic action videos collected from YouTube. It contains 13,320 videos from 101 action categories, with videos exhibiting large variations in camera motion, object appearance and pose, illumination conditions, and more.

For zero-shot image-text retrieval, we used:

- Flickr30k Plummer et al. (2015): A dataset containing 31,783 images collected from Flickr, each paired with 5 crowd-sourced captions. It focuses on describing the objects and actions in everyday scenes. The dataset is split into 29,783 training images, 1000 validation images, and 1000 test images.

- COCO Lin et al. (2014): A large-scale dataset for object detection, segmentation, and captioning, which we use for its image-caption pairs in the retrieval task. It features over 330,000 images, each with 5 captions. The dataset includes 80 object categories and instance segmentation masks, making it versatile for various computer vision tasks.

These datasets comprehensively evaluate a model's ability to perform zero-shot classification across various domains and its capacity for cross-modal retrieval. By using this diverse set of benchmarks, we can assess the generalization capabilities of our approach compared to existing CLIP models. We use Visually Descriptive Class-Wise prompts from Maniparambil et al. (2023) to enable the unimodal-text encoder in our DINOv2-ARL projector model to better identify the zero-shot classes of the downstream datasets.

### A.10.1 CONCEPT COVERAGE COLLECTION DATASETS

We use a few shot examples from 14 curated computer vision datasets to construct our Concept Image prototypes to curate the images from our uncurated data pool. The 14 curated datasets are described as follows.

- BirdSnap Berg et al. (2014): A fine-grained dataset consisting of 49,829 images of 500 North American bird species. The images are annotated with species labels, and the dataset is primarily used for species classification and fine-grained recognition tasks.

- Caltech101 Li et al. (2022): A dataset containing pictures of objects belonging to 101 categories, plus a background category. It includes about 40 to 800 images per category, with most categories having about 50 images. The dataset is known for its high intra-class variability.

- EuroSAT Helber et al. (2019): A satellite image dataset with 10 categories related to land use classification (e.g., forests, rivers, residential areas). It contains 27,000 labeled images, with 2700 images per class, widely used in remote sensing and geospatial tasks.

- FGVC Aircraft Maji et al. (2013): A fine-grained classification dataset with 10,000 images of 100 aircraft model variants from 70 manufacturers. It is used for distinguishing between visually similar objects in fine-grained recognition tasks.

- Flowers102 Nilsback & Zisserman (2008): A dataset containing 102 flower categories, commonly used for fine-grained classification tasks. It has a total of 8,189 images, with 40 to 258 images per category, and is organized into a training, validation, and test set.

- Food101 Bossard et al. (2014): A dataset containing 101,000 images of 101 food categories. Each category has 750 training images and 250 test images, commonly used for food classification and recognition tasks.

- GTSRB Stallkamp et al. (2012): The German Traffic Sign Recognition Benchmark dataset, containing over 50,000 images of 43 different traffic sign classes. It is designed for multi-class classification tasks in the context of traffic sign recognition.

- ImageNet Deng et al. (2009): A large-scale dataset with 1,000 object categories, widely used as a benchmark for image classification tasks. It contains over 1.2 million training images and 50,000 validation images, with each image labeled with one of 1,000 object classes.

- Oxford Pets Parkhi et al. (2012): A dataset of 7,349 images, containing 37 categories of pets (both cats and dogs). Each image is annotated with species and breed information, commonly used for image classification and segmentation tasks.

- RESISC45 Cheng et al. (2017): A dataset of remote sensing images used for scene classification, containing 31,500 images across 45 scene classes. Each class has 700 images with variations in resolution, scale, and orientation.

- Stanford Cars Krause et al. (2013): A dataset with 16,185 images of 196 car models, annotated by make, model, and year. The dataset is designed for fine-grained classification and recognition tasks of vehicles.

- Pascal VOC 2007 Everingham et al. (2015): A dataset for object detection, segmentation, and classification, containing 9,963 images of 20 object categories. It is widely used for benchmarking models in computer vision tasks.

- SUN397 Rouach et al. (2020): A large-scale scene understanding dataset with 397 categories and 108,754 images. It covers a wide range of environments, from natural to man-made scenes, commonly used for scene classification tasks.

- UCF101 Soomro et al. (2012): A video dataset consisting of 13,320 videos across 101 human action categories. It is widely used for action recognition tasks in video analysis and computer vision research.

