# OpenReview forum: "From Unimodal to Multimodal:Scaling up Projectors to Align Modalities"
_ICLR.cc/2025/Conference — ICLR 2025 Conference Withdrawn Submission_

### Official Review · Reviewer_MJ2t · 2024-10-20

**Soundness:** 2
**Presentation:** 3
**Contribution:** 3
**Rating:** 3
**Confidence:** 4

**Summary:**

This paper introduces an efficient multimodal alignment approach, where projectors are trained with a contrastive objective on top of frozen, pretrained unimodal encoders. The authors present extensive experiments related to multimodal alignment, data curation, and projector design. The resulting image-text model surpasses other baselines across several datasets.

**Strengths:**

* The paper explores a promising direction to leverage pretrained unimodal models for multimodal tasks.
* It includes extensive experiments and ablation studies.
* The final model achieves superior performance compared to other baselines across various datasets.

**Weaknesses:**

* The paper claims to outperform other CLIP baselines while being 20/65-fold more efficient. However, it relies on these same baselines to curate the datasets. If pretrained CLIP models are used, the cost of building those models should also be considered. Additionally, the cost of training the pretrained unimodal encoders (e.g., DINOv2, Roberta) should be accounted for, as CLIP models are trained from scratch. In general, the claims regarding efficiency seem inaccurate, and it is difficult to argue for efficiency under a fair comparison. Without the efficiency claim, the paper contribution becomes very limited.

* Important baselines seem to be missing. For example, finetuning a pretrained CLIP model on the curated dataset or training linear projectors on top of frozen CLIP encoders on the same dataset could could be valid baselines, especially when the paper leverages pretrained CLIP models.

* I think there is other measures for alignment, besides CKA that could be considered, such as cosine similarity, which has been shown to correlate with performance [1], or mutual nearest neighbor alignment [2]. Discussing these alternatives would strengthen the paper and provide better contextualization.

* Section 4.1 mentions “Unimodal Performance Does Not Reflect Alignment Quality”. This claim seems to contradict the results in the platonic framework. Any clarifications why could be the case in this setup?

[1] Shukor, Mustafa, and Matthieu Cord. "Implicit Multimodal Alignment: On the Generalization of Frozen LLMs to Multimodal Inputs." NeurIPS 2024.

[2] Huh, Minyoung, et al. "The platonic representation hypothesis." ICML 2024.

**Questions:**

* Please check weaknesses section.
* Did the authors try or have a plan to avoid using CLIP in their approach? this could strengthen the paper message

---

### Official Review · Reviewer_Gon4 · 2024-11-01

**Soundness:** 2
**Presentation:** 2
**Contribution:** 1
**Rating:** 3
**Confidence:** 3

**Summary:**

- Paper proposes a method for aligning pre-trained unimodal vision and language models using lightweight projection layers.
- The projection layers are trained leveraging inherent semantic similarities between the existing unimodal embedding spaces.
- The approach significantly reduces computational and data requirements. It achieves competitive performance on zero-shot classification and retrieval tasks.

**Strengths:**

- The proposed method achieves competitive performance with less compute and data compared to traditional multimodal training.
- The framework may be adapted to different scenarios, such as multilingual models and long-context retrieval.
- The experiments show good performance on various zero-shot classification and retrieval tasks.

**Weaknesses:**

- The idea of aligning via projection layers it not novel and is very common. The conclusion states "Our research introduces a paradigm shift in vision-language alignment." This seems like a major overstatement.
- The performance relies on the quality and coverage of the curated dataset. More analysis around the implication of this for broader generalization would be good.

**Questions:**

1. Why is this considered a paradigm shift?
2. What are the guidelines for the concept selection process? For example, how do you make sure the concepts are representative and unbiased?

---

### Official Review · Reviewer_Ntri · 2024-11-05

**Soundness:** 3
**Presentation:** 2
**Contribution:** 1
**Rating:** 3
**Confidence:** 4

**Summary:**

- The paper explores the use of unimodal encoders for contrastive training by using MLP projects on top of frozen encoders
- The paper proposes a method to select semantically similar encoders using Centered Kernel Alignment (CKA)
- The paper curates a dataset of image caption pairs for training the projectors using a contrastive loss
- The paper shows results on various tasks like 0-shot classification, retrieval, multi-lingual, localization, etc. and attains 76% top-1 on imagenet 0-shot

**Strengths:**

- The paper considers an interesting problem of creating CLIP style models from pretrained encoders
- The 0-shot localization, multi-lingual and long-text retrieval results are interesting
- There is a detailed appendix with more results and ablations

**Weaknesses:**

- The datasets are curated to include evaluations of interest, like SBU for Flickr evals, or ImageNet concepts for ImageNet “0-shot” which improves performance significantly as would be expected and Table 2 confirms this
- While the CKA metric shows some correlation with the text retrieval performance, the data doesn’t show that it’s a strong enough signal to choose an (image encoder, text encoder) pair. Overall this section seems like a very weak contribution in the paper. DINOv2’s main focus is linear classifier performance – it’s no surprise it’s the best model for CLIP finetuning if the encoder is frozen.
  - Fig. 1 shows a wide range of models with similar CLIP losses but varying CKA. Also, the primary determinant for both CKA and CLIP loss was the model size, so CKA might not really tell much about how “alignable” model pairs are.
  - Fig. 5 shows ConvNext get good retrieval performance (close to dinov2) with much lower CKA across all text encoders. Also, Fig. 5 doesn’t show all text encoders for each vision model family
  - There isn’t a comparison between frozen image encoder performance and retrieval performance.
- The plots in Fig. 1 are very comprehensive, but hard to learn much from. In Fig. 5 also it’s very hard to know which model pair a data point corresponds to.
- I don’t see enough contributions in the paper to warrant acceptance.
  - Data curation is important, but the datasets are very heavily curated for the evals.
  - When using comparable datasets (CC12M + CC3M + SBU) the paper gets 54% on ImageNet. LiT (table 13) attains 67% with much smaller models using just CC12M when using pretrained encoders but training the text encoder fully. This conveys to me that the LiT setup could be more efficient than just training projectors. There are no ablations / experiments around this. Also, LiT does talk about a similar problem, the paper claims it needs 256 TPU cores over 4 billion pairs, but there are experiments with less data.
- Table 1 / Section 4.2 don’t share details about the number of params in each layer (mlp vs. identity vs. token). Is it just that more parameters results in better performance since there’s more trainable parameters?  Why is the token projection layer skipped for text? Overall, why should we stop with a single token proj layer? Why should we freeze the encoders? There need to be many more experiments around these questions for researchers to know why this is a good setup to use. There is discussion around compute efficiency to motivate freezing the models, and that is a valid limitation, but it’s still not clear if this is the most compute optimal way to train the models. What if we added Lora layers to the encoders as well?

**Questions:**

- Please see weaknesses.

---

### Official Review · Reviewer_CysT · 2024-11-11

**Soundness:** 3
**Presentation:** 3
**Contribution:** 3
**Rating:** 6
**Confidence:** 4

**Summary:**

This paper proposed a systematic pipeline to identify unimodal encoders useful for multimodal transfer, transferring two image and text unimodal encoders into multimodal paired encoders by a stronger projector and high-quality dataset. The proposed model outperforms the original CLIP and other CLIP-like models, and the ablation is comprehensive.

**Strengths:**

1. The paper proposed an effective way (CKA) to measure how potentially good a uni-modal encoder would behave when transferring to multi-modal situations.
2. The paper proves that with a stronger projector and high-quality dataset, the unimodal encoders can better be transferred into multi-modality. The way to build high-quality dataset is interesting and insightful.
3. The result is impressive. The proposed method can outperform the original CLIP in some 0-shot image classification tasks, and also all presented retrieval tasks and most of the multi-lingual tasks.

**Weaknesses:**

1. A good property of CLIP is the effectiveness of scaling up, both in data and model scale. However, whether further scaling up the tunable parameter in the projector and high-quality training data (eg, from 20M to 50M, 100M, 1B) would also further improve the proposed method hasn't been studied.
2. Another important usage of CLIP-like model is to connect with LLM as a multimodal LLM. Whether the proposed model also have such capability may need to be verified.
3. Linear probing image classification as in original CLIP paper is also needed as additional experiments to verify the capability of proposed model.

**Questions:**

Please see the weaknesses.
Besides, I wonder if the high-quality training dataset will be published or not.

---

### Note · Authors · 2024-11-16

I have read and agree with the venue's withdrawal policy on behalf of myself and my co-authors.